# Amplification of chirality in surface-confined supramolecular bilayers

Hai Cao[1] & Steven De Feyter[1]

One of the most dramatic effects of supramolecular assembly is the generation of homochirality in near-racemic systems. It is normally infeasible though to flip the absolute chirality of a molecule. Here we rationalize this seemingly contradictory chiral amplification mechanism with a combined scanning tunneling microscopy (STM) and modeling study of surface-grown enantiomerically unbalanced supramolecular bilayers. We identify a chemical equilibrium between opposite but not mirror-image-related twisting molecular geometries of the pure enantiomer, and accordingly two competing aggregation pathways. The nonlinear chiral amplification effect in bilayers of near-racemic mixtures involves the biased adsorption and organization of the majority enantiomer, and the compliance of the minority enantiomer to adopt an energetically less favorable twisting molecular conformation and handed organization. By establishing a direct link between molecular building block architectures and chiral amplification effect, this study provides a general approach to gain insight into cooperative supramolecular assembly in mixed enantiomer systems.

[1] Division of Molecular Imaging and Photonics, Department of Chemistry, KU Leuven, Celestijnenlaan 200F, B3001 Leuven, Belgium. Correspondence and requests for materials should be addressed to H.C. (email: caohai@iccas.ac.cn) or to S.D.F (email: steven.defeyter@kuleuven.be)

Homochirality in life is an intriguing phenomenon. Several mechanisms have been postulated that might be at its origin or contributed to this fascinating reality[1,2]. Although the inversion of absolute molecular chirality is not possible for most chiral substances, there are ways to amplify chirality, given there is a minute enantiomeric imbalance in a racemate[3–6]. One particular promising avenue for the amplification of chirality is via cooperative supramolecular assembly[7–10], whereby a small enantiomeric excess at the molecular level is able to steer the aggregation of a near-racemic system toward homochiral assemblies. This nonlinear chiral amplification effect, known as majority rules[11], has increasingly become the subject of study in the last decade.

Mechanistic studies of the amplification of chirality thus far have predominantly looked at the effect of enantiomeric imbalances on the assembling of molecules. Based on the study of solution-borne one-dimensional (1D) systems, Meijer et al. linked the nonlinear chiral amplification effect to the association of the minority component to the handed superstructure of the majority enantiomer, at the expense of mismatching penalties[12–16]. While high-resolution local probe microscopies such as scanning tunneling microscopy (STM) can be applied to examine two-dimensional (2D) chiral networks at the molecular scale on surfaces[17–22], so far only few particular cases on chiral amplification

in mixed enantiomer systems have been reported[23–25], all of them focussing on monolayer systems. A boundary-driven amplification mechanism was revealed by Ernst and co-workers in a helicene system, where the chiral bias is created by the accumulation of excess enantiomer at the boundaries of enantiomorphous domains of the racemate[23]. A different pathway was proposed by Raval et al.[24] and later confirmed by Wan et al.[25], in which a slight excess of one enantiomer on the surface can suppress the adsorption and crystallization of the other one. While the chiral amplification effect has been shown to be related to structural properties of the molecular building blocks[13,14], a molecular-level description of the underlying driving forces that lead to the amplification of supramolecular chirality is lacking.

Here we take into account the conformational flexibility of molecules[26–30] and rationalize the nonlinear amplification effect with a combined STM and modeling study in supramolecular bilayers of near-racemic mixtures of alanine derivatives at the liquid–solid interface. We reveal opposite interconvertible twisting forms (Fig. 1a) and two competing aggregation pathways (Fig. 1b) of the pure enantiomers, and link the chiral amplification effects in supramolecular systems of the pure enantiomer (Fig. 1c) as well as of the non-racemic mixture (Fig. 1d) to the compliance of the minority conformer/enantiomer to match the preferred aggregate type of the majority component. By

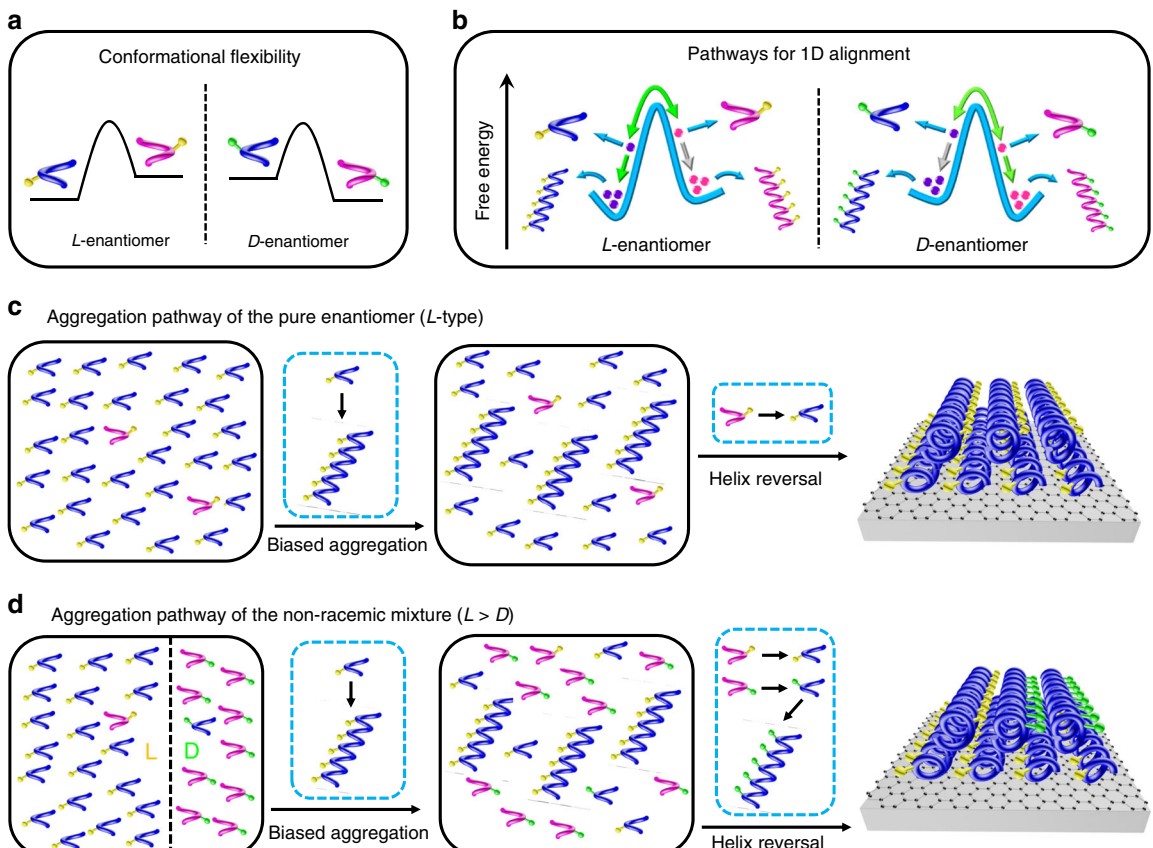

**Fig. 1** Pathway complexity in amplification of chirality. **a** Monomers of the two enantiomers—their absolute configuration is indicated by yellow (L-enantiomer) and green (D-enantiomer) dots—can exist in opposite twisting forms (magenta or blue) that are not mirror-image-related. **b** Two possible pathways of a pure enantiomer that are related to the formation of 1D aggregation of opposite organizational chirality, but only one route is feasible (indicated by green arrows) while the other one is predicted by modeling but not observed by STM observations, and therefore is called an inactive pathway (gray arrows). **c** Amplification of chirality in supramolecular bilayers of a pure enantiomer. The formation of homochiral bilayers of the pure enantiomer is initiated by the aggregation of the majority conformer, concomitant with the helix reversal of the minority conformer. **d** Aggregation pathway of an enantiomerically enriched mixture. The minority enantiomer tends to adopt the less-favored twisted conformation going hand in hand with it self-assembling according to the otherwise inactive aggregation pathway (see above), leading to its incorporation into the handed lattice of the majority enantiomer

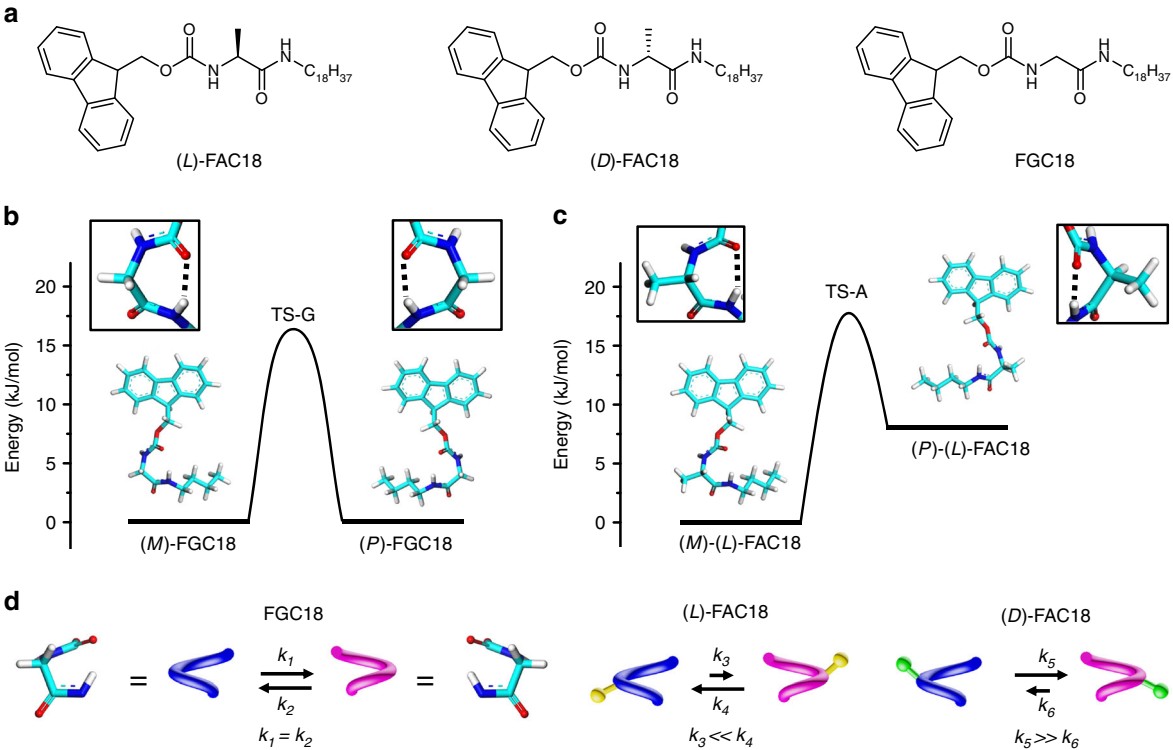

**Fig. 2** Transient molecular conformations. **a** Molecular structures of (L)-FAC18, (D)-FAC18, and FGC18. **b, c** DFT-optimized geometries and corresponding energy diagrams of FGC18 and (L)-FAC18, respectively. Insert boxes show the conformations of amino-acid segments. The black dashed lines indicate intramolecular hydrogen bonding. The transition-state structures are displayed in Supplementary Fig. 2. A C4 alkyl chain was used to reduce calculation time. **d** Schematic representation of the chemical equilibriums between $M$ (in blue) and $P$ (in magenta) twisting forms of FGC18, (L)-FAC18, and (D)-FAC18. The yellow and green pendent balls represent the methyl group at the chiral center of (L)-FAC18 and (D)-FAC18, respectively. $k_1$, $k_2$, $k_3$, $k_4$, $k_5$, and $k_6$ are the rate constants of helical reversal processes

rationalizing the chiral amplification mechanism at the molecular level, our findings are relevant for the understanding of cooperative supramolecular assembly in multiple-component systems.

## Results

**Conformational flexibility of monomers.** Derivatives of L-alanine and D-alanine, termed as (L)-FAC18 and (D)-FAC18, are used as the chiral monomers. Their achiral glycine counterpart, FGC18 (Fig. 2a), acts as a reference. The attachment of a fluorenyl-methyloxy-carbonyl (Fmoc) group at the N termius and a C18 alkyl chain at the C terminus of the amino acid implies a straightforward 1D supramolecular organization of these molecules on graphite via $\pi$–$\pi$, hydrogen bonding and hydrophobic interactions between the three parts of adjacent molecules. While the fluorene moiety is rigid and the alkyl chain typically prefers an extended geometry when adsorbed on graphite, the conformation of the amino-acid segment is more flexible (see Supplementary Fig. 1). These three molecules can exist in many conformations at the monomeric state. Shown in Fig. 2b, c are the optimized geometries of FGC18 and (L)-FAC18 as revealed by density functional theory (DFT) calculations. The two geometries of a molecule—equal in energy for FGC18 and differing by 8.3 kJ/mol for the enantiomers—stand apart from each other in the way the amino-acid segment is twisted. We denote the opposite twisting forms as $P$-type and $M$-type helicity. The $P$- and $M$-forms of FGC18 are mirror-image-related, whereas the added methyl group at the chiral carbon breaks such mirror image relationship and creates a bias toward $M$-type helicity for (L)-FAC18 and $P$-type helicity for (D)-FAC18. The $P$- and $M$-forms of FGC18 coexist with equal probability, but vast majority of (L)-FAC18 is $M$-type (the population in the gas phase is

estimated to be 96.5% at room temperature according to the Boltzmann distribution) while $P$-type helicity is preferred for (D)-FAC18. The barrier for helix reversal amounts to 16.9 kJ/mol for FGC18 and 18.5 kJ/mol for (L)-FAC18, implying the possibility of interconversion between two conformers. Figure 2d illustrates the chemical equilibriums between $P$-helicity and $M$-helicity of FGC18, (L)-FAC18, and (D)-FAC18 at the monomeric state.

**Predictive assembly pathways of the pure enantiomers.** For steric reasons, growth of 1D paralleled structures of the pure enantiomer should be favored for molecules of the same helicity. To probe the effect of molecular chirality and helicity on the aggregation pathways, we conducted molecular mechanism (MM) calculations on the 1D periodic arrays of (L)-FAC18, (D)-FAC18, and FGC18. In view of the mirror image relationship between $(P)$-FGC18 and $(M)$-FGC18, between $(P)$-(L)-FAC18 and $(M)$-(D)-FAC18 and between $(M)$-(L)-FAC18 and $(P)$-(D)-FAC18, only the energy profiles of $(M)$-(L)-FAC18, $(P)$-(L)-FAC18, and $(M)$-FGC18 as a function of the intermolecular distance are selected and displayed in Fig. 3a and Supplementary Figs. 3, 4 to illustrate the aggregation patterns of the three molecules. When the intermolecular distance drops from 0.54 to 0.44 nm, two minima can be observed in all the energy profiles (Fig. 3a). It appears that the energy profile of less-favored $(P)$-(L)-FAC18 is very similar to that of FGC18, and in both cases the lowest energy point corresponds to an intermolecular distance of ~0.45 nm. $(M)$-(L)-FAC18, on the other hand, possesses a lower energy minimum above 0.48 nm but a higher energy minimum around 0.45 nm as compared to the $P$-conformer. For all compounds, a similar transition in molecular alignment, from oblique above 0.48 nm (I and II in Fig. 3b) to near-rectangular around 0.45 nm

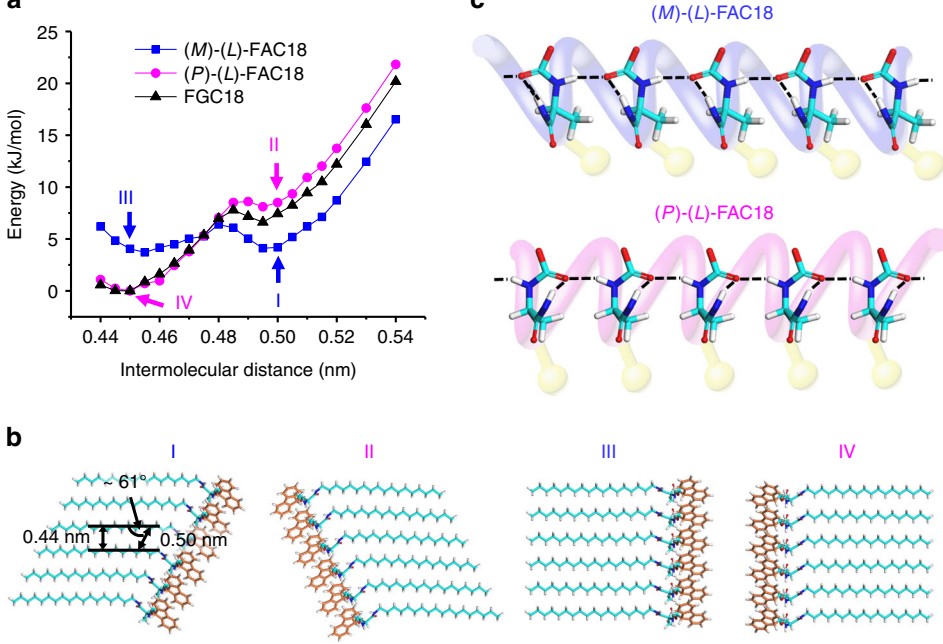

**Fig. 3** Competing aggregation pathways. **a** Energy profiles of the 1D arrays of FGC18 and P- and M-forms of (L)-FAC18, as a function of the intermolecular distance. The energy minimum of (L)-FAC18 and FGC18 arrays are set to zero for comparison. **b** Four supramolecular organizations along the energy profiles of (L)-FAC18, corresponding to 1D slanting arrays of (I) M- and (II) P-conformer with a intermolecular distance of 0.50 nm, and 1D near-rectangular arrays (III) M- and (IV) P-conformer with a regular spacing of 0.45 nm. The intermolecular distance (0.50 nm), the spacing between alkyl chains (~0.44 nm) and the oblique angle (~61°) of organization I are indicated. The fluorene group of (L)-FAC18 is displayed in orange for clarity. I–IV of (D)-FAC18 and FGC18 are defined in the same way and displayed in Supplementary Fig. 4. **c** Illustration and "helix" representation of 1D alignment of M- and P-forms of (L)-FAC18. The intermolecular and intramolecular hydrogen bonding are indicated by dashed lines

(III and IV in Fig. 3b), was revealed. Modeling predicts that above 0.48 nm, the M-conformers of (L)-FAC18, (D)-FAC18, and FGC18 all prefer a right-slanting arrangement while the P-conformers of these three molecules all favor a left-slanting arrangement (Fig. 3b and Supplementary Fig. 4). In other words, this modeling approach indicates that (L)-FAC18, (D)-FAC18, and FGC18 are all able to form stable 1D oblique arrays, of which the inclination directions, to the left or to the right, are decided exclusively by the molecular helicity (Fig. 3c). The absolute molecular chirality, on the other hand, is decisive in determining which inclination direction is energetically more favorable.

**Supramolecular bilayers of the pure enantiomers.** Despite the variability in molecular conformation and 1D organization, as indicated by the molecular modeling approach, experimentally only parallel-arranged oblique 1D arrays were observed for enantiopure (L)-FAC18 and (D)-FAC18 upon depositing their 1-phenyloctane solutions ($c = 0.2$ mM) onto the surface of highly ordered pyrolytic graphite (HOPG), as revealed by STM (Fig. 4a–e). The dimensions of a domain of this lamellar structure can extend to a few hundred nanometers but occasionally with the presence of defects and voids (Fig. 4a and Supplementary Fig. 5, 6). From high-resolution STM images of the surface structures of (L)-FAC18 and (D)-FAC18 (Fig. 4b, c), narrow bright strands that are separated by wide dim stripes can be clearly identified. We ascribe the dim stripes to parallel aligned alkyl chains and the bright protrusions to the fluorene groups and the amino-acid moieties. The orientation of the alkyl chains was revealed to follow one of the main symmetry axes of the underlying graphite lattice, while the lamellar direction follows another. For both enantiomers, the long vector $b$ of a unit cell amounts to $3.6 \pm 0.1$ nm. The short vector $a$, however, is too short to be

precisely measured, but an average value of 0.5 nm can be determined on the basis of cross-section analysis. The angle, $\gamma$, between the two vectors is $70 \pm 1°$. We define the surface organizational chirality of these enantiomorphous structures by the smallest angle $\theta$ (~10°) between one of the symmetry axes of graphite and the unit cell vector $b$ as clockwise (CW) and counterclockwise (CCW) handednesses for (L)-FAC18 and (D)-FAC18, respectively. An alternative and visually more attractive approach is to consider the angle between unit cell vector $a$ and the alkyl chains, leading to the same identifications of handedness.

In consideration of the correlation between molecular helicity and inclination direction of the 1D array revealed by theoretical calculations (Fig. 3a–c), helicity of (L)-FAC18 and (D)-FAC18 in the experimentally observed surface-confined networks can be determined as M-type and P-type, respectively, i.e., adopting the via theoretical modeling established preferred twisting geometries of pure enantiomers. However, the many bright protrusions in a narrow bright strand in the STM images do not conform to head-to-tail arranged monolayer networks, as four levels of topography can be identified from a close-up image such as in Fig. 4d. In addition, dislocations in the lamellar structures reveal the presence of narrow less bright strands running parallel to or in the extension of the apparent narrow bright strands (Fig. 4e). Such defects can be observed at different locations and in different experiments. Given that the brightness of features in the STM images relates to the topography—the brighter the protrusion the larger the apparent height[31]—the distinction in contrast of two isolated strands at nearly the same location is indicative of the fact that the lamellar structure is actually composed of two layers of molecules. To form a bilayer, the two layers of molecules can either be parallel displaced or oppositely oriented, but only the latter agrees well with the transition in

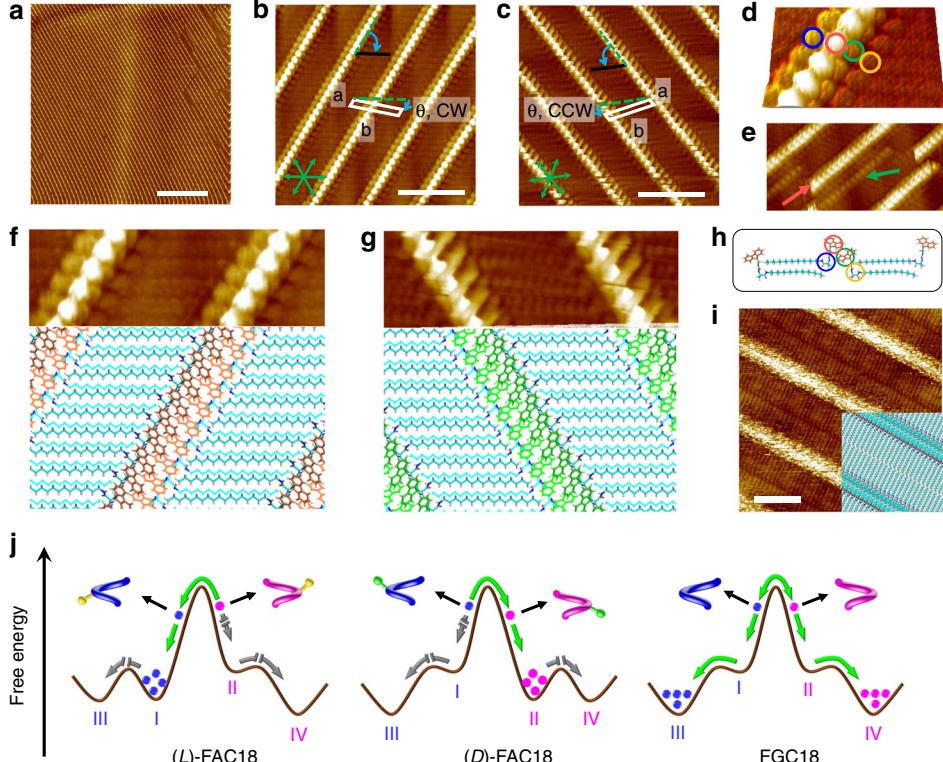

**Fig. 4** Supramolecular bilayers of the pure enantiomers. **a** Large-scale STM image of the lamellar network of (L)-FAC18 ($I_{set}$ = 200 pA, $V_{bias}$ = −800 mV). **b, c** High-resolution STM images of the enantiomorphous structures of (L)-FAC18 ($I_{set}$ = 200 pA, $V_{bias}$ = −800 mV) and (D)-FAC18 ($I_{set}$ = 200 pA, $V_{bias}$ = −600 mV), respectively. The green arrows and dashed lines indicate the orientation of main symmetry axes of the underlying graphite lattice. The white parallelograms outline the unit cells of surface structures, while the black lines indicate the orientations of alkyl chains. CW and CCW organizational chirality is defined by the angle between the HOPG reference axis and the unit cell vector $b$. **d** 3D view revealing four types of bright protrusions (outlined by circles in different colors). **e** STM image of a defect where a bright strand is separated into two isolated ones (indicated by red and green arrows). **f, g** Small-scale STM images and energy-minimized CW and CCW structures of the bilayer arrangement of (L)-FAC18 and (D)-FAC18, respectively, with unit cell dimensions (nm) 0.50 × 3.55, $\gamma$ = 70°. The top and bottom layers are displayed in stick and line representations, respectively. **h** Side view of the bilayer. The circles outline the moieties that contribute to the different bright features in **d**; **i** STM image and structural model of the monolayer structure of FGC18 ($I_{set}$ = 10 pA, $V_{bias}$ = −600 mV). **j** Self-assembly energy landscapes of (L)-FAC18, (D)-FAC18, and FGC18. The green arrows indicate the active pathways involved in the formation of surface structures, while the gray arrows show the pathways that have been predicted by simulations but are not observed by STM, i.e. the so-called inactive pathways. Scale bars: **a** 50 nm; **b, c, i** 5 nm

bright features, as illustrated by the optimized model structures shown in Fig. 4f–h, where the bilayer arrangement is stabilized by hydrophobic interactions and π–π interactions between two layers. Different kinds of defects can therefore be interpreted as the mismatch and dislocation in the bilayer arrangement (see Supplementary Figs. 5, 6). Further, the potential energy surface of such bilayer structure obtained by MM simulations exhibits a minimum with the dimensions (nm) of 0.50 × 3.55 (see Supplementary Fig. 7), well in accordance with the experimental observations.

While P- and M-forms of a single enantiomer of FAC18 coexist at the molecularly dissolved state and they are in principle able to self-assemble into structures of opposite handedness, the observation of homogeneous CW and CCW structures of the pure enantiomers is indicative of the fact that a slight energy difference in the stability of conformers at the molecular level is significantly amplified upon supramolecular assembly. Therefore, there is no experimentally observable evidence for the helix reversal of an enantiomer, from the preferred conformer to the less stable one of opposite helicity, and the related assembly of such less stable helix conformer into the by theoretical modeling predicted mirror image oblique or near-rectangular supramolecular organizations.

In other words, these self-assembly processes of the less stable helix conformers are inactive (Fig. 4j).

Stimulated by the similarity between the energy profiles of FGC18 and the less-favored helix conformers of FAC18, we as well studied experimentally the self-assembly of FGC18 at the 1-phenyloctane/HOPG interface. We expected that the self-assembly patterns of this achiral compound would bring insight into the structural aspects of the so-far inactive and therefore not-experimentally observed yet self-assembly route(s) and organization of the less stable conformers of FAC18. Compared to the enantiomers, achiral analog FGC18 forms a different monolayer phase (Fig. 4i), which reflects a near-rectangular alignment, that is, the global minimum state along the energy landscape (Fig. 4j). Hence, the above-predicted oblique alignment of FGC18 represents only a shallow local minimum. It can therefore be envisaged that the spontaneous organization of the less-preferred conformer of an enantiomer of FAC18—if present on the surface—should lead to the formation of a near-rectangular network, which is different from the aggregation preference of the energetically favored conformer. In other words, the appearance of a near-rectangular organization of FAC18, not yet experimentally observed, can be considered as

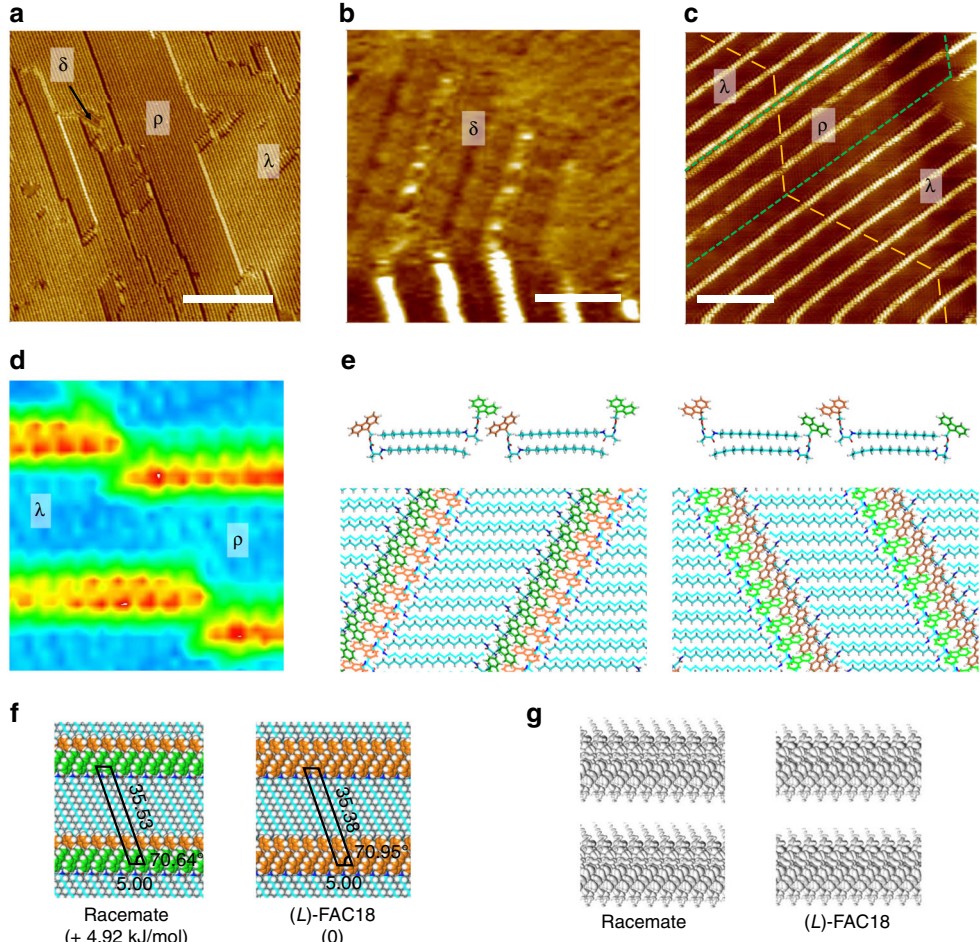

**Fig. 5** Various phases of the racemate. **a** Large-scale STM image of the lamellar structure formed from the equimolar mixture of (L)-FAC18 and (D)-FAC18, where three types of domains (λ, ρ, and δ) can be identified according to their brightness and pattern. $I_{set}$ = 150 pA, $V_{bias}$ = −800 mV. **b** STM image showing a small patch of the δ phase ($I_{set}$ = 20 pA, $V_{bias}$ = −600 mV). **c** STM image showing the coexistence of λ and ρ domains. $I_{set}$ = 200 pA, $V_{bias}$ = −600 mV. The yellow lines indicate the orientation of alkyl chains (see Supplementary Fig. 9 for enlarged image), while the green dashed lines outline the domain boundaries. **d** Close-up image showing transition in contrast from a λ domain to a ρ domain. **e** Structural models for the CW and CCW bilayer structure of the racemate, respectively, with dimensions (nm) 0.50 × 3.55, $\gamma$ = 70°. The fluorene groups of (L)-FAC18 and (D)-FAC18 are displayed in orange and green, respectively. The top and bottom layers are displayed in stick and line representations, respectively. **f** A comparison between the lattice parameters (in Å) and the energy (displayed in parenthesis) of DFT-optimized CW bilayers of (L)-FAC18 and the racemate. **g** HOMO to HOMO-3 isosurfaces of the CW bilayer structures of the racemate and (L)-FAC18. Scale bars: **a** 100 nm; **b** 5 nm; **c** 10 nm

an indication of the presence of the less-preferred conformer on the surface.

**Different self-assembled domains of the racemate.** We then moved on to investigate the cooperative aggregation of the racemic mixture of both FAC18 enantiomers on graphite. At the first sight, the lamellar structures formed by the racemate (Fig. 5a and Supplementary Fig. 8) seem to be similar to those of the pure enantiomers, but few aspects cannot be explained by the 2D segregated aggregation of both enantiomers. First, a noticeable difference in topology was observed, which we identified as λ, ρ, and δ domains (Fig. 5a–c). δ domains only appear in very small patches (Fig. 5b). Compared with ρ domains, λ domains are slightly brighter (Fig. 5c), and a transition from λ to ρ may occur in a strand, whereas the connection always appears as a break or a kink (Fig. 5d). The ρ domains show always the same handedness but with undetermined organizational chirality, that is, either CCW or CW in a continuous domain. CW and CCW structures can be observed in adjacent rows but only in the λ domains

(Fig. 5c and Supplementary Fig. 9). The occasionally observed δ phase of the racemate was not revealed for the pure enantiomers (Fig. 5b and Supplementary Fig. 10). It resembles the self-assembled structures of FGC18, and is likewise a monolayer structure formed by head-to-head arranged molecular arrays, which incidentally confirms the bilayer nature of the λ and ρ domains. As discussed above, oblique and near-rectangular arrangements are preferred for the aggregation of the energetically favored and the less stable conformers, respectively (Fig. 4j), and the emergence of δ phase—a pattern that should in principle only be attained via inactive pathways—therefore provides a hint of the participation of less-preferred conformers of FAC18 in the surface assembly.

It is known that the aggregation of a racemate may result in the formation of a conglomerate, in which each domain contains only one enantiomer, or the formation of a racemic compound containing equal amounts of the two enantiomers in a domain[2,32]. These two possibilities for the crystallization of a racemate are likely the cause of the discrepancy between λ and ρ domains. To prove this argument, we preformed calculations to

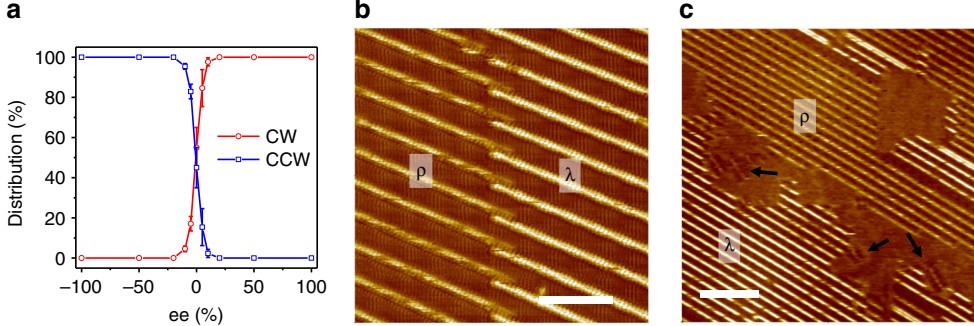

**Fig. 6** Nonlinear amplification of chirality in enantiomerically unbalanced supramolecular bilayers. **a** Distribution of the CW and CCW structures of FAC18 on graphite as a function of the enantiomeric excess (ee) in the solution phase, where ee = (L − D)/(L + D). The δ phase was not taken into account. The error bars in **a** were calculated as a s.d. of at least three measurements. **b** STM image of the CW structure of a (L)-FAC18:(D)-FAC18 = 3:2 mixture, showing the coexistence of λ and ρ domains. $I_{set}$ = 200 pA, $V_{bias}$ = −600 mV. **c** STM image showing the submonolayer coverage of the racemate. Black arrows indicate the monolayer δ phase of the racemate. $I_{set}$ = 20 pA, $V_{bias}$ = −600 mV. Scale bars: **b** 10 nm; **c** 20 nm

obtain the bilayer structure of the racemate. However, it is not feasible to incorporate directly the preferred (M)-(L)-FAC18 conformer into a CCW domain that is favored by (D)-FAC18, or the (P)-(D)-FAC18 conformer into a CW domain. One of the two enantiomers must adjust its configuration or conformation to match the handed arrangement of the other one to form a CW or CCW supramolecular bilayer of the racemate. Inversion of the absolute molecular chirality, i.e. L and D handedness, can be ruled out, leaving helix reversal the only option to implement the association of one enantiomer into a superstructure having organizational chirality opposite to its preference. As such, optimum bilayer structures of the racemate were obtained with one enantiomer in the bottom layer but the other one in the top layer, as illustrated in Fig. 5e and Supplementary Fig. 7. Such CW and CCW structures of the racemate are formed on condition that the L- and D-enantiomers in the two layers possess the same helicity and that one of both enantiomers assembles following a formerly discussed inactive aggregation pathway, the one giving rise to an oblique arrangement. Supramolecular bilayer of the racemate possesses similar lattice parameters as compared to that of the pure enantiomer, but is ~5 kJ/mol less favored (Fig. 5f and Supplementary Fig. 7).

Since STM topography is a manifestation of the local density of states of adsorbed molecules, additional clues for the helix reversal of one enantiomer may lie in the transition in the topological contrast, due to variations to the molecular conformation and arrangement. Small differences in highest occupied molecular orbital were revealed by DFT calculations, between the opposite conformers of an enantiomer (see Supplementary Fig. 2) and between the supramolecular bilayers of the racemate and the pure enantiomers (Fig. 5g). In view of the simulated STM appearance, and taking into account the zigzag pattern in λ domains and the uniform organizational chirality in a ρ domain, we therefore ascribe the λ phase to a racemic compound and the ρ phase to a racemic conglomerate. As such, mismatch between a λ domain and a ρ domain can be interpreted as a result of the minute difference between lattice parameters of the two. The δ phase that resembles the monolayer structure of achiral FGC18, on the other hand, is the energetically more stable product of the inactive aggregation pathway.

It should be noted that the occurrence of helix reversal is possible in both layers with nearly the same penalty cost if the surface is not taken into account (see Supplementary Fig. 7). However, it is more likely that the helical reversal is triggered by preformed surface structures, as such an energetically unfavorable process is unlikely to occur spontaneously.

**Nonlinear amplification of chirality in non-racemic mixtures**. Adsorption of the racemate gives rise to an almost equal distribution of the CW and CCW domains. When the system goes from a (L)-FAC18:(D)-FAC18 = 50:50 situation to near-racemic to enantiopure, a nonlinear chiral amplification effect was revealed. The overall organizational chirality, namely the coverage of CW and CCW domains, is dependent on the enantiomeric excess (ee), as shown in Fig. 6a (see Supplementary Table 1 for details). At ee = 5%, a significant change to the chirality was observed, as the surface was covered with 85% CW domains but 15% CCW domains only. Further increasing the content of (L)-FAC18 to 55% in the mixture resulted in an almost homochiral situation as 98% of the lamellar networks show CW handedness. When ee ≥ 20%, no CCW structures can be observed any more. We noticed that λ domain corresponding to the networks of the racemate and ρ domain of the excess enantiomer coexist on surface when homochirality was reached in the enantiomerically unbalanced bilayers (Fig. 6b and Supplementary Fig. 11). While the dominance of CW structures is associated with excess (L)-FAC18, the added (D)-FAC18 biases the aggregation into CCW structures. This nonlinear amplification, therefore, is a manifestation of the majority rules effect.

We then sought to pinpoint the origin of chiral amplification. An enantiomeric imbalance-induced dominance of the majority enantiomer in the surface-confined monolayer has been reported by Raval and co-workers[24]. Also Ernst et al. have shown that a small excess of one enantiomer in a double-layer system may lead to the adsorption of the majority enantiomer in the bottom layer and the minority enantiomer in the top layer[33]. The former has been ascribed to an entropy-driven chiral ordering mechanism, while the latter occurs only in condition of the coverage exceeds one monolayer. In our case, we assure that there is no selective adsorption of one enantiomer over the other one on surface (see Supplementary Discussion for arguments). In addition, it usually takes more than an hour to reach a near-full surface coverage, and neither the formation of bilayer structures nor the occurrence of helix reversal requires full monolayer coverage. Using the monolayer δ phase as a reference, we observed the formation of bilayer λ and ρ domains immediately after the deposition of solution (Fig. 6c). That is to say, the occurrence of non-spontaneous helix reversal of minority enantiomer is not driven by competing adsorption, as there is enough space for the adsorption and segregated organization of the minority enantiomer at a sub-surface coverage. We invoke the mathematical and kinetic models proposed by Meijer et al.[15,34,35], and propose a competing aggregation mechanism in which the enantiomeric

imbalance creates a bias toward the aggregation of the majority enantiomer. Then assemblies of the majority enantiomer act as seeds to guide the growth of the minority enantiomer (Fig. 1). As such, no full surface coverage is needed for helix reversal, and the magnitude of helix reversal energy barrier and penalty is decisive in determining the efficiency of chiral amplification. It can be speculated that changing the molecular structure and temperature may alter the chemical equilibrium and therewith affect the nonlinear effect[13,14].

In conclusion, we have rationalized a nonlinear chiral amplification mechanism at the molecular scale, via investigating surface-supported supramolecular bilayers of near-racemic mixtures by means of STM and modeling. By taking into account the chemical equilibrium between opposite twisting molecular geometries of the enantiomers and the existence of an inactive aggregation pathway, we have elucidated how a chiral molecule can be well incorporated into the handed lattice of its mirror image enantiomer without changing its absolute molecular chirality. Our results imply that the enantiomeric imbalance is necessary to create a bias toward the aggregation of the majority enantiomer therewith to facilitate the helical reversal of the minority enantiomer, whereas the chemical equilibrium between transient molecular forms and the competition between different aggregation pathways play decisive roles in the amplification of chirality. While conformational flexibility is not limited to amino-acid derivatives (see Supplementary Fig. 12 for a few examples), but applies to a large amount of other organic species, we envisage that such findings can pave the way to new researches into the pathway complexity in multi-component supramolecular systems.

## Methods

**Materials**. The three compounds used in this study were chemically synthesized by a one-step condensation reaction of Fmoc-protected amino acid and octadecylamine[36]. Details for the synthesis of (L)-FAC18: Fmoc-L-Ala-OH (hydrate, 2 g, 6.4 mmol, TCI); octadecylamine (1.7 g, 6.4 mmol, Alfa); EDC.HCl (3 g, 15.6 mmol, GL Biochem); and HOBt (anhydrous, 2.1 g, 15.6 mmol, GL Biochem) were mixed in dichloromethane (50 mL) in a 100 mL flask and stirred at room temperature for about 1 week. Subsequently, yellow product was collected upon the removal of solvent and purified via silica column chromatography (gradient elution with $CH_3OH/CH_2Cl_2/CHCl_3 = 1/80/20$ to 3/80/20). The target product was obtained as a fine white solid (3.2 g, 87% yield). (D)-FAC18 and FGC18 were synthesized in a similar way. All the three compounds have been characterized by $^1H$ nuclear magnetic resonance, matrix-assisted laser desorption ionization-time of flight mass spectrometry, and elemental analyses[36]. 1-phenyloctane (98%, Sigma-Aldrich, used as received) was used as solvent to dissolve FGC18, (L)-FAC18, and (D)-FAC18. The concentration used in all the experiments in this study is $2 \times 10^{-4}$ M.

**STM measurements**. All STM experiments were performed at room temperature (20–23 °C) using a PicoSPM (Molecular Imaging, now Keysight) machine operating in constant-current mode with the tip immersed in the supernatant liquid. STM tips were prepared by mechanical cutting from Pt/Ir wire (80%/20%, diameter 0.2 mm). Prior to imaging, a drop of the solution was applied onto a freshly cleaved surface of HOPG (grade ZYB, Advanced Ceramics Inc., Cleveland, USA). Typical parameters for the imaging of molecules on graphite are $V_{bias} = -600$ to $-800$ mV and $I_{set} = 100$ to 200 pA, where $V_{bias}$ and $I_{set}$ are sample bias and tunneling current, respectively. For analysis purposes, recording of a monolayer or bilayer image on HOPG was followed by consecutive imaging the graphite substrate underneath. This was done under the same experimental conditions but by lowering the substrate bias (typically $V_{bias} = -1$ mV) and increasing the tunneling current (typical $I_{set} = 800$ to 900 pA). From the atomically resolved STM image of HOPG, the graphite symmetry axes can be easily determined. The images were corrected for drift via the Scanning Probe Image Processor software (Image Metrology ApS), using the graphite lattice, allowing a more accurate unit cell determination. The error bars of unit cell parameters were calculated as a s.d. of more than three measurements. All the STM images are low-pass filtered.

**Molecular modeling**. First, the possible conformations of FGC18 and (L)-FAC18 were searched by using a simulated quench method with Compass force field (Compass-II)[37] available in Material Studio. The quench starts with some conceived molecular conformations, using some available crystal structures of the Fmoc-derived molecules as references[38–40]. A cutoff of 1.85 nm was applied for electrostatic and van der Waals interactions. A unit of 20 ps-long molecular dynamics simulations are conducted in the NVT ensemble (constant number of particles (N), volume (V) and temperature (T)) at 300 K with a time step of 1 fs. Quenching every 100 steps yields 200 structures, of which a few of the most stable conformations were selected for further DFT calculations with Gaussian 09[41], using the B97D functional with a 6–31++G(d,p) basis set. The transition states between different conformations were searched with the Synchronous Transit-Guided Quasi-Newton method. All the calculations were performed in vacuum. A comparison of different methods is shown in Supplementary Table 2.

The 1D organizations of FGC18 and FAC18 were also optimized using the Compass-II force field. DFT-optimized molecules were placed at a fixed intermolecular distance of 0.50 nm and then subjected to simulated quench process at 300 K for 30 ps with a time step of 1 fs. Only the most stable trapped and β-sheet-like arrangements of FGC18 and FAC18 were collected and analyzed for comparison (see Supplementary Fig. 3). The starting geometries at varying intermolecular distances were constructed on the basis of optimum arrangements obtained by quenching. The minima along the energy profiles were further verified using simulated quench process, revealing nearly the same optimum arrangements and energy.

The bilayer organizations of FAC18 were obtained by placing two oppositely oriented molecules in a given unit cell in the way that observed by STM. First, the bilayer arrangement was probed by several simulated quench processes (30 ps at 300 K with a time step of 1 fs) at a fixed vector $b = 3.7$ nm but $a = 0.48, 0.49, 0.50, 0.51$, and 0.52 nm, respectively. Geometries in unit cells of varying parameters were further constructed on the basis of optimum arrangements obtained by quenching and optimized. For each potential surface, 9 values of the vector $a$ and 11 values of the vector $b$ were selected for calculation, that is, 99 data points for each graph. The optimized structure and corresponding energy were collected and analyzed.

Optimized bilayer structures of the pure enantiomer and the racemate were further calculated by DFT, using the Castep module available in Material Studio 2017 in a generalized gradient approximation. A fixed vacuum layer larger than 30 Å was applied. The gradient-corrected Perdew Burke Enzerhof exchange-correlation functional is employed together with an energy cutoff of 600 eV and $3 \times 1 \times 1$ k-points. The van de Waals interactions are described by a semi-empirical Grimme DFT-D2 approach[42,43]. The ionic core is represented by ultrasoft pseudopotentials. All the results are based on non-spin polarization calculations with density mixing scheme.

**Data availability**. All relevant data that support the findings of this study are available from the corresponding authors upon request.

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

## Acknowledgements

We thank Dr. Kunal Mali (KU Leuven) for fruitful discussions. This work is in part supported by the Fund of Scientific Research Flanders (FWO), KU Leuven—Internal Funds, FWO under EOS 30489208, and the Hercules Foundation. The research leading to these results has also received funding from the European Research Council under the European Union's Seventh Framework Programme (FP7/2007–2013)/ERC Grant Agreement No. 340324.

## Author contributions

H.C. and S.D.F. conceived and designed the concepts. H.C. synthesized the glycine and alanine derivatives, acquired and analyzed the STM data, and performed the theoretical simulations. H.C. and S.D.F. co-wrote the paper.

## Additional information

**Competing interests:** The authors declare no competing interests.

