## [Peer Review File · Nature Communications]

Editorial Note: Parts of this Peer Review File have been redacted as indicated as we could not obtain permission to publish the reports of Reviewer 2

Reviewers' comments:

Reviewer #1 (Remarks to the Author):

The paper describes experiments and supporting theory related to experiments on chirality switching in bilayers of molecules on a graphite surface. This is a comprehensive study and the experimental results and supporting theory appear reliable. However the paper comes across as slightly specialised and as someone with interests in surface assembly but rather peripheral to the sub-field of surface chirality I found it difficult to understand the main points of the paper.

Nevertheless I don't question the potential significance of the work and perhaps the authors could overcome some of the issues by revising the paper. In particular I found the 'scene-setting' comments towards the end of page 3, including the diagram Scheme 1 rather difficult to follow. What is the reference to glycine? I think in Scheme 1 the authors are trying to get over the idea that a chiral species could include a unit which can undergo conformational changes and pack together either as CW or CCW helix and that both chiral partners could form either CW or CCW helices. I did spend a long time looking at Scheme 1 - a problem is that the helix components don't look very three dimensional so this point is rather confusing. On a related point I didn't understand what was meant by describing one of the pathways as 'silent'.

The discussion of Fig 2 seems clearer - even here there is potential for confusion due to the many acronyms for the different conformations, chiralities and molecules. Also I couldn't follow the arguments at the bottom of p11 - again related to silent aggregation. On Figure 3 and 4 I can see that there is something interesting going on when both enantiomers are deposited together but can't follow the arguments as to how the origin of the different phases is related to the structures with differing chirality.

The conformational freedom available in this molecule certainly leads to interesting effects but the discussion becomes difficult to follow due in part to the many variations and complexity. Perhaps the authors could revise the paper in a way which makes these arguments in a more accessible manner.

Reviewer #2 (Remarks to the Author):

[redacted]

Reviewer #3 (Remarks to the Author):

Cao and de Feyter present an in-depth study about the self-assembly of a molecule with a complex set of degrees of freedom involving stereo symmetry but also conformational flexibility. This provides a very remarkable milestone in that there is unprecedented insight into the interplay of chirality and conformational changes upon supramolecular organization. This interplay is related to an important molecule and is at least by this reviewer expected to be of general importance in the self organization of molecular material also far beyond the currently investigated system of the chosen Alanine (Amino acid) derivatives.

This is a very carefully prepared study and report, both theory and experiment are presented in a very convincing way. I particularly like that the authors chose the achiral analogon of Alanine as the reference material to verify their hypotheses.

The topic of research and the presentation of the results are very well suited for Nature Communications. In contrast, however, I feel that the authors should take more care to explain their results and the conclusions that can be drawn to the readership of Nature Communications from a broad disciplinary background. The current introductory paragraph is rather long and more historic than solution geared after briefly introducing into the scientific challenge.

N.B. the final lines of the Abstract read very nice, but the language is a bit on the specialists side. What I like here is that it starts with 'generation of homochirality in racemic systems' unlike the first line of the article which spans a bridge between life's homochirality (which is a specific feature of synthesis in biomolecular chemistry) and chiral amplification (which if I am right is more a feature of scientific investigation related to supramolecular assemblies and not to biomolecular synthesis) and is less common in nature.

In my view the 'majority' rules effect is here modified such that the majority forces the minority to change the conformation, to adopt a less favoured conformational state in order to show compliance.

This, I repeat myself, deserves to be explained and evidenced with pride of the authors and should be accepted after mandatory revisions for publication in Nature Communications.

Further comments:

Given that the brightness of features in the STM images relates to the topography -- the brighter the the protrusion, the larger the apparent height – the distinction in contrast of two isolated strands at nearly

With the long history of STM studies on longer chain aliphatic derivatives – liquid crystals, thiols and more, the referencing could be improved to evidence the identification of 'chemical' (i.e. HOMO / LUMO) and topographic contrast. I just point at one manuscript here: Y. Qian et al., Langmuir 2003, 19, 6056-6065

P16, line 307: ..., while the latTer occurs only IN condition of the coverage exceeding one ML. (or similar).

I agree with the authors that the brightness in the STM images HERE relates to the topography, but this is not generally the case and should be discussed a bit more carefully, possibly with respect to the expected electronic and topographic contrast from the alkane chains and the fluorenyl group.

I find the second part of the title not so well justified: " A probe into the complexity of supramolecular copolymerization". It could be that 'supramolecular polymerisation' (It seems like it is a technical term in a significant part of the community, but I admit I was not familiar with it

and am not sure if everybody is) is just used as a synonym for 'layer formation' or 'formation/packing of a supramolecular assembly' it may imply that in some cases of polymerization self-organisation will be a prerequisite of polymerization i.e. the formation of a covalent or coordination polymer. In the absence of a bio-catalyst (enzyme) such processes occur at elevated temperatures and in presence of a bio-catalyst it is more the site specificity of the active site. The author's work has enough to offer on its own, this implication is not needed, for sure not in the title while a statement along this line could be added to the conclusion.

Another suggestion: In the conclusion it may be worth to extrapolate on what molecular building block architectures e.g. other amino acids and beyond, could provide a similar set of properties i.e. an energy for conformational changes that could reduce the 'defect energy' of a racemic crystallization to a pseudo-homochiral one. Also interesting would be a discussion of the energy required for the conformational change as this energy is expected to shift the 'majority rules' guideline to a 'xx% rules' guideline. This could be mentioned together with some statement on how such further insight could be gained in experiment and theory – actually the theory contained in the article here should be able to predict and discuss this issue and I don't recall that I saw it.

Leuven, 21th May 2018

Dear Reviewer,

We very much appreciate your constructive criticisms, and have modified the manuscript significantly to address your concerns.

Thank you once again for your interest in our work and for helping us in improving the manuscript. As you will see, we have made a serious effort to make the manuscript easily accessible to a broad readership.

Modifications to the manuscript can be traced in the “for review only” file indicating via track changes the modifications made.

Sincerely yours,

Steven De Feyter and Hai Cao

REVIEWER COMMENTS AND AUTHOR RESPONSE:

Reviewer #1 (Remarks to the Author):

1. The paper describes experiments and supporting theory related to experiments on chirality switching in bilayers of molecules on a graphite surface. This is a comprehensive study and the experimental results and supporting theory appear reliable. However the paper comes across as slightly specialised and as someone with interests in surface assembly but rather peripheral to the sub-field of surface chirality I found it difficult to understand the main points of the paper. Nevertheless I don't question the potential significance of the work and perhaps the authors could overcome some of the issues by revising the paper. In particular I found the 'scene-setting' comments towards the end of page 3, including the diagram Scheme 1 rather difficult to follow.

Author reply:

We thank the reviewer for his/her evaluation of our manuscript.

We have made substantial changes to both the text and the figures, to clarify the main points, and to make the manuscript more accessible to a wide audience. In particular, Scheme 1 has been revised completely to show how conformational flexibility at the molecular level evolves into diverse self-assembly pathways and eventually leads to the generation of homochirality in near-racemic systems.

2. What is the reference to glycine?

Author reply:

The glycine derivative acts as an important reference in this study:

- 1) Investigations on the glycine derivative are primarily intended to verify the reliability of pathway complexity revealed by theoretical calculations.
- 2) Considering the similarity between alanine and glycine, investigation on the glycine analogue can act as a good example to show the dramatic effect of methyl side chain on the conformation flexibility of molecular building blocks and therewith the aggregation pathways.
- 3) For each enantiomer, there are two potential aggregation pathways predicted by the calculation, corresponding to the 1D alignment of two helical conformers. For each pathway, there are two possible supramolecular arrangements, i.e. inclined and near-rectangular. The preferred supramolecular arrangement of the preferred conformer is revealed by STM observations. In contrast, we do not know the assembly behaviour of the less preferred conformer in the absence of experimental observations. However, modelling shows that the energy landscapes of the less preferred conformer of the *L*-enantiomer (*P*-type conformer) and *D*-enantiomer (*M*-type conformer) are nearly identical to those of the achiral glycine counterpart. We therefore use the glycine derivative as a reference to predict the preference for spontaneous assembly of the less favoured conformer, if at all it would occur.

3. I think in Scheme 1 the authors are trying to get over the idea that a chiral species could include a unit which can undergo conformational changes and pack together either as CW or CCW helix and that both chiral partners could form either CW or CCW helices. I did spend a long time looking at Scheme 1 - a problem is that the helix components don't look very three dimensional so this point is rather confusing. On a related point I didn't understand what was meant by describing one of the pathways as 'silent'.

Author reply:

We have changed the scheme. The 'silent' pathway refers to the aggregation pathway which is predicted by theoretical modelling to take place for the energetically less preferred conformer and which in principle could lead to the global minimum state on the energy landscape, but which is not observed by STM for enantiopure systems. We have specified and clarified the meaning of 'silent' pathway – now called “inactive” pathway – in the revised manuscript in the caption of Scheme 1.

4. The discussion of Fig 2 seems clearer - even here there is potential for confusion due to the many acronyms for the different conformations, chiralities and molecules. Also I couldn't follow the arguments at the bottom of p11 - again related to silent aggregation.

Author reply:

As compared to inclined bilayer structures of the pure enantiomers, the achiral analogue FGC18 forms a different monolayer phase, which reflects a near-rectangular alignment and implies that the via theoretical modelling predicted oblique alignment of FGC18 is only a shallow local minimum on the energy landscape. Given the similarity between the energy profiles of FGC18 and the less favoured conformers of FAC18, we envisage that a similar monolayer structure – a phase that is different from the observed bilayer structure of the pure enantiomers – would be obtained upon the spontaneous organization of the less preferred conformers, that is, via a silent (inactive) aggregation pathway. Or looked at it in a different way, the experimental observation of small patches of monolayer δ phase – a near-rectangular network that should in principle only be attained via the 'inactive' pathway – in mixed enantiomer systems is indicative of the presence of the less preferred conformer

on the surface. Helical reversal is therefore considered as the most likely reason for the appearance of different phases in the racemic and non-racemic mixtures.

We rewrote the discussion related to the 'silent' ('inactive') pathway.

5. On Figure 3 and 4 I can see that there is something interesting going on when both enantiomers are deposited together but can't follow the arguments as to how the origin of the different phases is related to the structures with differing chirality.

Author reply:

We completely rewrote this section, and developed our arguments in the following way:

Experimental observations:

- Surface structure of a pure enantiomer shows always the same contrast in a STM image. While *L*-enantiomer forms exclusively CW bilayers on the surface, only CCW bilayers are observed for the *D*- enantiomer.
- In contrast to the enantiopure systems, three different types of domains, denoted as λ , ρ and δ , can be identified when a mixture containing both enantiomers is deposited, according to their supramolecular organisation and STM contrast.
 - All rows in a ρ domain show the same handedness, which could be CW or CCW.
 - A λ domain may contain rows of CW, CCW or both chiralities.
 - Small patches of monolayer δ phase are never observed for the pure enantiomers.

Principle of racemate crystallization:

- While the chirality/handedness of the supramolecular pattern of an enantiomer is unique (CW or CCW), the self-assembly of a racemate may in principle lead to the formation of two different types of aggregates:

- 1) a racemic conglomerate in which a single domain contains only one enantiomer, therewith the organizational chirality of a domain is either CW or CCW, depending on which enantiomer is adsorbed,
- 2) a racemic compound, in which both enantiomers are present in the same domain, hence the handedness of the supramolecular pattern in a domain is not determined, and depends on how two enantiomers are organized.

These two typical ways a racemic mixture crystalizes are likely the reason why different phases are observed for the racemate.

Modelling:

- Aggregation preference of the preferred conformer of an enantiomer: CW structure of the *L*-enantiomer can be linked to the 1D oblique arrangement of the preferred *M*-conformer with a regular intermolecular spacing of 0.50 nm. CCW structure of the *D*-enantiomer, on the other hand, is formed by the preferred *P*-conformer. Near-rectangular arrangements of these preferred (*M*)-(*L*)-FAC18 and (*P*)-(*D*)-FAC18 conformers cannot be obtained, as they are formed at a shorter intermolecular distance (~0.45 nm) but are nearly equal in energy with their oblique counterparts. There is a barrier in between these two organizations on the energy landscapes (see Fig. 2a and 3j).
- Aggregation preference of the less preferred conformer of an enantiomer (“inactive pathway”): the energy profile of the less preferred conformer of an enantiomer is nearly identical to its glycine counterpart. While the near-rectangular organization of the glycine derivative observed by STM is indicative of the fact that the oblique arrangements represent a local minimum which is too shallow to trap the aggregation (see Fig. 2a and 3j), a similar monolayer phase with near-rectangular lattice can be expected from spontaneous aggregation of the less preferred conformer of an enantiomer, which explains the appearance of the δ phase of the racemate. In other words, the presence of the δ phase is indicative of the involvement of the less preferred conformers in aggregation of the racemate.
- CW and CCW bilayers of the racemate: since the preferred molecular conformation and supramolecular organization of two enantiomers are determined to be mirror

image related, it is not possible to directly incorporate one enantiomer to the preferred handed lattice of the other one. Therefore, the most feasible way is the compliance of one of both enantiomers to adopt an energetically less favourable twisting molecular conformation and handed organization. As such, CW and CCW bilayers can be obtained with one enantiomer in the bottom layer but the other one in the top layer whereby the *L*- and *D*-enantiomers in the two layers possess the same helicity. Theoretical calculations show that a CW (or CCW) supramolecular bilayer of the racemate possesses nearly the same lattice parameters as those of the pure enantiomer, but is ~5 kJ/mol less favoured. Also a small difference in molecular arrangements (Fig. S6) and HOMO (Fig. 4g) is revealed between the supramolecular bilayers of the racemate and the pure enantiomers.

Combining all of the above leads to this conclusion:

- λ phase: racemic compound
- ρ phase: racemic conglomerate.

6. The conformational freedom available in this molecule certainly leads to interesting effects but the discussion becomes difficult to follow due in part to the many variations and complexity. Perhaps the authors could the revise the paper in a way which makes these arguments in a more accessible manner.

Author reply:

We thank the reviewer for the good suggestions. We have made substantial changes to the manuscript, in particular to the scheme and the figures to make sure the paper is easier to read.

Reviewer #2 (Remarks to the Author):

[redacted]

Reviewer #3 (Remarks to the Author):

1. Cao and de Feyter present an in-depth study about the self-assembly of a molecule with a complex set of degrees of freedom involving stereo symmetry but also conformational flexibility. This provides a very remarkable milestone in that there is unprecedented insight into the interplay of chirality and conformational changes upon supramolecular organization. This interplay is related to an important molecule and is at least by this reviewer expected to be of general importance in the self organization of molecular material also far beyond the currently investigated system of the chosen Alanine (Amino acid) derivatives. This is a very carefully prepared study and report, both theory and experiment are presented in a very convincing way. I particularly like that the authors chose the achiral analogon of Alanine as the reference material to verify their hypotheses.

Author reply:

We thank the reviewer for his/her positive evaluation of our manuscript.

2. The topic of research and the presentation of the results are very well suited for Nature Communications. In contrast, however, I feel that the authors should take more care to explain their results and the conclusions that can be drawn to the readership of Nature Communications from a broad disciplinary background. The current introductory paragraph is rather long and more historic than solution geared after briefly introducing into the scientific challenge.

Author reply:

The introductory paragraphs have been modified to emphasize the novelty of this work. While all the studies thus far focus on the impact of enantiomeric imbalance on the organizing of molecules, in this study we take into account the chemical equilibrium between opposite twisting molecular geometries of the enantiomers and the existence of competing aggregation pathways, and present a molecular level description of the underlying driving forces that lead to the amplification of supramolecular chirality.

3. N.B. the final lines of the Abstract read very nice, but the language is a bit on the specialists side. What I like here is that it starts with ‘generation of homochirality in racemic systems’ unlike the first line of the article which spans a bridge between life’s homochirality (which is a specific feature of synthesis in biomolecular chemistry) and chiral amplification (which if I am right is more a feature of scientific investigation related

to supramolecular assemblies and not to biomolecular synthesis) and is less common in nature.

Author reply:

We have changed it from ‘Such findings highlight the importance of transient molecular conformations, diverse aggregation pathways and competing nucleation and growth processes on the amplification of chirality and supramolecular copolymerization in general.’ to ‘By establishing a direct link between molecular building block architectures and chiral amplification effect, this study provides a new approach to gain insight into cooperative supramolecular assembly in mixed enantiomer systems.’

4. In my view the ‘majority’ rules effect is here modified such that the majority forces the minority to change the conformation, to adopt a less favoured conformational state in order to show compliance. This, I repeat myself, deserves to be explained and evidenced with pride of the authors and should be accepted after mandatory revisions for publication in Nature Communications.

Author reply:

We have modified Scheme 1 to highlight the compliance of minority conformer/enantiomer in chiral amplification.

Further comments:

5. Given that the brightness of features in the STM images relates to the topography -- the brighter the the protrusion, the larger the apparent height – the distinction in contrast of two isolated strands at nearly With the long history of STM studies on longer chain aliphatic derivatives – liquid crystals, thiols and more, the referencing could be improved to evidence the identification of ‘chemical’ (i.e. HOMO / LUMO) and topographic contrast. I just point at one manuscript here: Y. Qian et al., Langmuir 2003, 19, 6056-6065

Author reply:

We have added the reference to support our statement.

6. P16, line 307: ..., while the latTer occurs only IN condition of the coverage exceeding one ML. (or similar).

Author reply:

The typo has been corrected in the revised version.

7. I agree with the authors that the brightness in the STM images HERE relates to the topography, but this is not generally the case and should be discussed a bit more carefully, possibly with respect to the expected electronic and topographic contrast from the alkane chains and the fluorenyl group.

Author reply:

We have compared the supramolecular bilayers of the racemate and the pure enantiomer by means of DFT calculations. Subtle variations in lattice parameters and an energy difference of ~5 kJ/mol were revealed. Since negative bias was applied in this study, therefore isosurface plots of the HOMO to HOMO-3 of the racemate and the pure enantiomers are presented to illustrate the transition in topographic contrast.

8. I find the second part of the title not so well justified: “ A probe into the complexity of supramolecular copolymerization”. It could be that ‘supramolecular polymerisation’ (It seems like it is a technical term in a significant part of the community, but I admit I was not familiar with it and am not sure if everybody is) is just used as a synonym for ‘layer formation’ or ‘formation/packing of a supramolecular assembly’ it may imply that in some cases of polymerization self-organisation will be a prerequisite of polymerization i.e. the formation of a covalent or coordination polymer. In the absence of a bio-catalyst (enzyme) such processes occur at elevated temperatures and in presence of a bio-catalyst it is more the site specificity of the active site. The author’s work has enough to offer on its own, this

implication is not needed, for sure not in the title while a statement along this line could be added to the conclusion.

Author reply:

The terms ‘supramolecular polymerization’ and ‘supramolecular copolymerization’ have both been widely used to describe the formation of non-covalent assemblies in solution, but are less frequently used in surface science. We have changed the second part of the title, from “A probe into the complexity of supramolecular copolymerization” to “A probe into the complexity of cooperative supramolecular assembly” to avoid the misunderstandings.

9. Another suggestion: In the conclusion it may be worth to extrapolate on what molecular building block architectures e.g. other amino acids and beyond, could provide a similar set of properties i.e. an energy for conformational changes that could reduce the ‘defect energy’ of a racemic crystallization to a pseudo-homochiral one. Also interesting would be a discussion of the energy required for the conformational change as this energy is expected to shift the ‘majority rules’ guideline to a ‘xx% rules’ guideline. This could be mentioned together with some statement on how such further insight could be gained in experiment and theory – actually the theory contained in the article here should be able to predict and discuss this issue and I don’t recall that I saw it.

Author reply:

We thank the reviewer for his/her good suggestions.

We believe that the conformational flexibility of amino acid derivatives is a general fact. We have compared the opposite twisting forms of a few amino acids (valine, leucine, isoleucine and phenylalanine) analogues with DFT calculations, revealing small energy differences in all these cases.

Valine

Leucine

Isoleucine

Phenylalanine

That is to say, similar set of properties can also be expected from those amino acids on condition that supramolecular bilayers could also be formed by those molecules. But we think that it calls for our further experimental and theoretical investigations to shift the ‘majority rules’ guideline to a ‘xx% rules’ guideline, as the type of side chain may have a dramatic effect on the way the molecules organize, and therewith affect the amplification of chirality.

REVIEWERS' COMMENTS:

Reviewer #1 (Remarks to the Author):

It is clear that the authors have made significant efforts to improve this paper but I unfortunately still find it very difficult to understand to the extent that I don't think I can provide an authoritative review.

Reviewer #3 (Remarks to the Author):

2nd Referee report for Nature Communication submission, "Amplification of chirality in surface-confined supramolecular bilayers: ..." by H. Co and S. de Feyter.

All three reviewers mention aspects in favour of a publication of this manuscript in Nature Comm, i.e. 'This is a comprehensive study and the experimental results and supporting theory appear reliable' (reviewer 1), 'I feel the work could be important but I am unsure that I understand the experimental protocol' (reviewer 2) and '... This provides a remarkable milestone in that there is unpercedented insight into the interplay of chirality and conformation changes upon supramolecular organization. This is a very carefully prepared study and report, both theory and experiment are presented in a very convincing way.' (reviewer 3).

Reviewer 1 (like reviewer 3): raised the issue of the 'science setting' comments being found only at the end of the manuscript. – this has been taken care of by a significant reorganization of the introduction, the scheme 1 and the discussion.

Reviewer 1 also raised that the scope of the glycerine reference is not clarified sufficiently. – this has been responded to and taken up in the revisions in the manuscript.

Reviewer 1 mentioned difficulties to understand scheme 1 – this has been improved in the new version of this scheme.

Reviewer 1 suggested improvements with regard to the discussion of the terminology of a 'silent' pathway: -- authors have significantly revised this discussion and removed the confusing terminology 'silent'.

Reviewer 1 further suggests to improve the the discussion of the self-assembly of racemic mixtures presented in Fig. 3 and 4. – authors have completely rewritten this section.

Last but not least Reviewer 1 asks for a revision of the storyline (again together with the other reviewers). – authors have complied with this request in the opinion of the author of these lines.

XX XXXXXX

Reviewer 2 (in some similarity to Reviewer 1 in his 5th point) raises the issue of enantiomer identification in STM. – authors have responded to both reviewers concerning this point and revised the relevant sections of the manuscript.

Reviewer 2 raises the – valid – point of selective adsorption mechanisms changing the enantiomeric excess in the adsorbed state from the value provided via the solution. The authors replied adequately, but in my humble opinion they could add a small phrase stating their 'assumption' i.e. 'we don't assume selective adsorption of one enantiomer over the other'.

\ Comment by this reviewer – such a selectivity can be ruled out from a symmetry consideration unless there is a significant excess of adsorption sites of one chirality over the other on the surface to be considered. To the best of my knowledge this is not the case on the terraces of the HOPG used in this study.

Reviewer 2 asks another valid question i.e. about the arguments towards or against helix reversal. -- I like the three arguments the authors provided here so much that I would motivate them to put as much as possible into the SI of the manuscript – and a reference in the main manuscript.

Last reviewer 2 raises an ambiguity with the word 'nucleation' which has a more specific meaning in crystallization. – the authors modified the terminology and used 'biased aggregation' now.

Finally, reviewer 3 is too tired to itemize his own responses, but is ready to state that he finds the author's responses and corrective actions suitable as they are delivered by the authors.

More specifically this reviewer advises the editor to accept the manuscript, but again advises the authors to take up some of the nice arguments i.e. about the conformational flexibility of aminoacides into the SI – or provide a more elaborate study in a follow up article.

Reviewer #4 (Remarks to the Author):

I think the paper presents an interesting combined experimental and theoretical study, which proposes a novel mechanism of chiral amplification of near racemic molecular ensemble. I'm not sure that the presented data convincingly proof the scenario, but I think the present data do not 100% support the claims. On the other hand, this paper can stimulate further investigation of chiral amplification promoted by the mechanism by other groups. As this it can stimulate further development and new discoveries in the filed. As this I recommend this paper for publication in Nature Communication.

REVIEWERS' COMMENTS:

Reviewer #1 (Remarks to the Author):

It is clear that the authors have made significant efforts to improve this paper but I unfortunately still find it very difficult to understand to the extent that I don't think I can provide an authoritative review.

Author reply: we thank the reviewer for his/her evaluation of our manuscript.

Reviewer #3 (Remarks to the Author):

2nd Referee report for Nature Communication submission, “Amplification of chirality in surface-confined supramolecular bilayers: ...” by H. Co and S. de Feyter. All three reviewers mention aspects in favour of a publication of this manuscript in Nature Comm, i.e. ‘This is a comprehensive study and the experimental results and supporting theory appear reliable’ (reviewer 1), ‘I feel the work could be important but I am unsure that I understand the experimental protocol’ (reviewer 2) and ‘... This provides a remarkable milestone in that there is unprecedented insight into the interplay of chirality and conformation changes upon supramolecular organization. This is a very carefully prepared study and report, both theory and experiment are presented in a very convincing way.’ (reviewer 3).

Reviewer 1 (like reviewer 3): raised the issue of the ‘science setting’ comments being found only at the end of the manuscript. – this has been taken care of by a significant reorganization of the introduction, the scheme 1 and the discussion.

Reviewer 1 also raised that the scope of the glycerine reference is not clarified sufficiently. – this has been responded to and taken up in the revisions in the manuscript. Reviewer 1 mentioned difficulties to understand scheme 1 – this has been improved in the new version of this scheme.

Reviewer 1 suggested improvements with regard to the discussion of the terminology of a ‘silent’ pathway: -- authors have significantly revised this discussion and removed the confusing terminology ‘silent’.

Reviewer 1 further suggests to improve the the discussion of the self-assembly of racemic mixtures presented in Fig. 3 and 4. – authors have completely rewritten this section. Last but not least Reviewer 1 asks for a revision of the storyline (again together with the other reviewers). – authors have complied with this request in the opinion of the author of these lines.

Author reply: we thank the reviewer 2 for his/her evaluation of our manuscript, and really appreciate the reviewer 3 for his/her interest in our work and for helping us in improving the manuscript.

Reviewer 2 (in some similarity to Reviewer 1 in his 5th point) raises the issue of enantiomer identification in STM. – authors have responded to both reviewers concerning this point and revised the relevant sections of the manuscript.

Reviewer 2 raises the – valid – point of selective adsorption mechanisms changing the enantiomeric excess in the adsorbed state from the value provided via the solution. The authors replied adequately, but in my humble opinion they could add a small phrase stating their ‘assumption’ i.e. ‘we don’t assume selective adsorption of one enantiomer over the other’.

Author reply: We thank the reviewer for the kind suggestion. We have added a few sentences in the main manuscript to state this assumption and provided Supplementary Discussion to support the claim.

Comment by this reviewer – such a selectivity can be ruled out from a symmetry consideration unless there is a significant excess of adsorption sites of one chirality over the other on the surface to be considered. To the best of my knowledge this is not the case on the terraces of the HOPG used in this study.

Author reply: we totally agree with the Reviewer’s points. Typically on metal surfaces, the under-coordinated atoms of kinked and stepped surface structures serve as active sites for the adsorption – in some particular cases the enantioselective adsorption – of organic species. But often the role of step edges of HOPG in molecular adsorption and organization is not taken into consideration, as the difference in molecular adsorption at the step edges and on the terraces of HOPG surface is often not pronounced.

We thank the reviewer for pointing this out. We have included it in Supplementary Discussion to rule out the possibility of selective adsorption of one enantiomer over the other one at the liquid/HOPG interface.

Reviewer 2 asks another valid question i.e. about the arguments towards or against helix reversal. -- I like the three arguments the authors provided here so much that I would motivate them to put as much as possible into the SI of the manuscript – and a reference in the main manuscript.

Author reply: we thank the reviewer for the suggestion. Argument 3 has already been discussed in the revised main manuscript. Further, we have added argument 2 in SI as

Supplementary Fig. 1 to show the conformational flexibility of amino acid segment and explain why we consider helix reversal the most likely reason for chiral amplification. We have also used argument 1 as Supplementary Discussion to support our claim that helix reversal rather than selective adsorption of one enantiomer over another is responsible for the amplification of chirality in enantiomerically unbalanced supramolecular bilayers.

Last reviewer 2 raises an ambiguity with the word ‘nucleation’ which has a more specific meaning in crystallization. – the authors modified the terminology and used ‘biased aggregation’ now. Finally, reviewer 3 is too tired to itemize his own responses, but is ready to state that he finds the author’s responses and corrective actions suitable as they are delivered by the authors.

More specifically this reviewer advises the editor to accept the manuscript, but again advises the authors to take up some of the nice arguments i.e. about the conformational flexibility of aminoacides into the SI – or provide a more elaborate study in a follow up article.

Author reply: we thank the reviewer again for carefully reading of our manuscript, looking over our response to all the comments and providing valuable advices.

We have made a few changes to SI according to the reviewer’s advices to support our claims and to show the implication of our findings in other supramolecular systems.

Reviewer #4 (Remarks to the Author):

I think the paper presents an interesting combined experimental and theoretical study, which proposes a novel mechanism of chiral amplification of near racemic molecular ensemble. I'm not sure that the presented data convincingly proof the scenario, but I think the present data do not 100% support the claims. On the other hand, this paper can stimulate further investigation of chiral amplification promoted by the mechanism by other groups. As this it can stimulate further development and new discoveries in the filed. As this I recommend this paper for publication in Nature Communication.

Author reply: we thank the reviewer for his/her evaluation of our manuscript, and the recommendation of accepting this work.